# Portable XRF Quick-Scan Mapping for Potential Toxic Elements Pollutants in Sustainable Urban Drainage Systems: A Methodological Approach

**Guri Venvik** [1],* and **Floris C. Boogaard** [2,3]

1   Geological Survey of Norway, P.O. Box 6315 Torgarden, 7491 Trondheim, Norway
2   NoorderRuimte, Centre of Applied Research and Innovation on Area Development, Hanze University of Applied Sciences, Zernikeplein 7, P.O. Box 3037, 9701 DA Groningen, The Netherlands; Floris@noorderruimte.nl
3   Deltares, Daltonlaan 600, 3584 BK, 85467 3508 AL Utrecht, The Netherlands
*   Correspondence: guri.venvik@ngu.no; Tel.: +47-7390-4313

**Abstract:** Sustainable urban drainage systems (SuDS) such as swales are designed to collect, store and infiltrate a large amount of surface runoff water during heavy rainfall. Stormwater is known to transport pollutants, such as particle-bound Potential Toxic Elements (PTE), which are known to often accumulate in the topsoil. A portable XRF instrument (pXRF) is used to provide in situ spatial characterization of soil pollutants, specifically lead (Pb), zink (Zn) and copper (Cu). The method uses pXRF measurements of PTE along profiles with set intervals (1 m) to cover the swale with cross-sections, across the inlet, the deepest point and the outlet. Soil samples are collected, and the In-Situ measurements are verified by the results from laboratory analyses. Stormwater is here shown to be the transporting media for the pollutants, so it is of importance to investigate areas most prone to flooding and infiltration. This quick scan method is time and cost-efficient, easy to execute and the results are comparable to any known (inter)national threshold criteria for polluted soils. The results are of great importance for all stakeholders in cities that are involved in climate adaptation and implementing green infrastructure in urban areas. However, too little is still known about the long-term functioning of the soil-based SuDS facilities.

**Keywords:** portable X-ray fluorescence spectrometer (pXRF); Potential Toxic Elements (PTE); lead (Pb), zinc (Zn); copper (Cu); topsoil; sustainable urban drainage systems; SuDS; LID; BMPs; WSUD; GI; SCMs

## 1. Introduction

In urban and densely populated areas, surface runoff can carry material residue produced by daily human activity and has been identified as an important pathway for pollutants that enter receiving water bodies [1,2]. Sustainable urban drainage systems (SuDS, also called green infrastructure (GI), best management practices (BMP), low impact development (LID), water sensitive urban design (WSUD), ecosystem-based adaptation (EbA), nature-based solutions (NBS) and others [3]) are constructed to receive, store and infiltrate surface water to restore the groundwater balance and to remove pollutants, such as lead, zinc and copper [4]. An increased pollutant load in urban stormwater degrades water quality, therefore knowledge of the characteristics of the pollutants is needed—vital knowledge that

can be incorporated into management and maintenance [5]. With climate change, a higher proportion of rainfall will become surface runoff [6], which in turn will result in increased peak flood discharges and subsequently degraded water quality [7].

As Tedoldi et al. [8,9] point out, few studies have carried out systematic mapping of the horizontal distribution of pollutants in SuDS. Studies by Jones and Davis [10] and Tedoldi et al. [8,9], show that the concentration of Potential Toxic Elements (PTE) are in the uppermost 0–12 cm of the soil in SuDS and decrease with depth, down to 90–100 m [8,10]. Investigation of SuDS constructions such as swales (Figure 1) by means of laboratory analysis of soil samples is costly and time-consuming. Systematic, large scale investigation of environmental-technical rainwater facilities has not been conducted due to the high cost of soil analysis. Past research indicates that contamination is restricted to the upper 10 to 30 cm of soil [9]. In the quick scan method, a portable X-ray fluorescence spectrometer (pXRF) measures a range of elements, including PTE. Portable XRF measurements provide an established method for analyzing metals and other elements [11–16], and in cases where low detection levels are not required, time-consuming and costly laboratory analyses could be minimized to control samples.

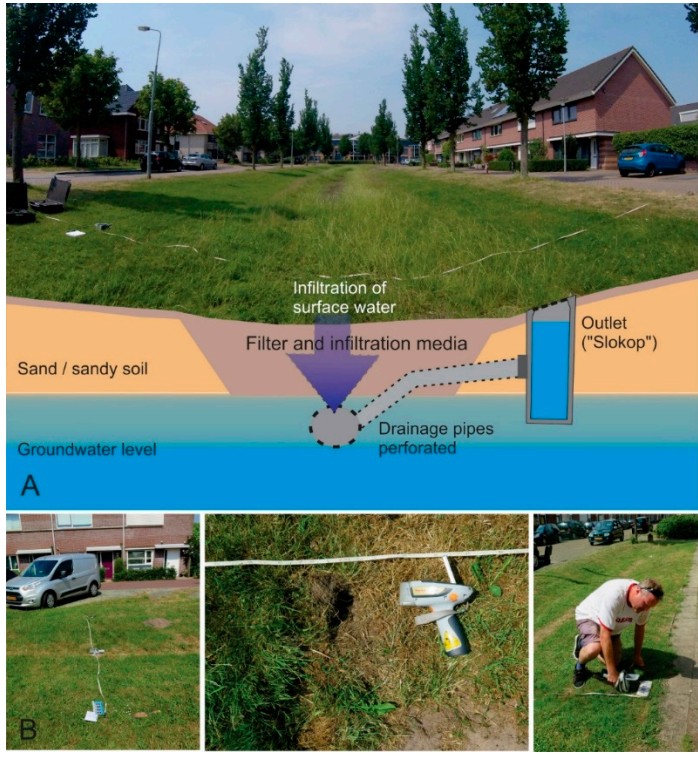

**Figure 1.** (**A**): Principle sketch of a swale. (**B**): Demonstration of in situ pXRF measurements at every 1 m along a profile across the swale. For the measuring results to be comparable with lab results for soil samples, the vegetation is removed to measure directly on the topsoil (0–3 cm). The instrument is pointed on the topsoil and each measurement is read for 60 s.

This paper proposes a methodology for mapping PTE, including lead (Pb), zinc (Zn) and copper (Cu), in situ in topsoil in SuDS. The study was conducted in the Netherlands where SuDS has been used for over a decade and is part of a national study [17]. A pXRF device has been used that can give immediate results in the field. In general, in situ analysis of the degree of pollution in the topsoil may ensure quick follow-up actions, like soil remediation. In situ pXRF measurements provide quick and cost-efficient analyses for PTE mapping.

The cost estimate for implementing SuDS usually includes installation but not management, monitoring or maintenance. Therefore, the analytical results are of great importance for all stakeholders in cities that are involved in the implementation and maintenance of green infrastructure for climate

adaptation [18]. Since swales (Figure 1) are the most common construction built for storing stormwater and infiltrating surface water in the Netherlands, more knowledge is needed on the long-term pollution levels in the soil. Therefore, this field-based study in the Netherlands aims to establish a methodology using in situ measurements by a pXRF instrument that are quick and cost-efficient.

## 2. Materials and Methods

The proposed approach was exploited for mapping SuDS used swales as pilot facilities (Figure 1). In situ pXRF measurements of topsoil (0–3 cm) were performed. Soil samples were collected at the same locality (also 0–3 cm) for lab analysis and quality control. Swales vary in shape and design from small swales along roads in residential areas (1–2 m wide) to large swales (5–7 m wide). Figure 1 shows the principles of a swale, commonly a depression in the ground with grass vegetation, an inlet for stormwater and outlet for surplus water. Further, swales are constructed with layers of filtrating media according to Woods Ballard et al. [2]. The three pilot locations presented here and more than 250 other Dutch swale locations are mapped in the open source webtool www.climatescan.org [19]. In the webtool the general characteristics of the sample area, photos and description were gathered at each location, where this research is available with description, images and videos.

This study focuses on the PTE lead (Pb), zinc (Zn) and copper (Cu) due to their prevalence in stormwater and their toxicity at high concentrations [5]. The threshold values for these PTE in soils, by Dutch regulations, are provided in Table 1.

**Table 1.** Dutch threshold values for pollutants in soil [20].

| Metals | National Background Concentration ppm (mg/kg) | Target Value ppm (mg/kg) | Intervention Value ppm (mg/kg) |
|---|---|---|---|
| Lead (Pb) | 85 | 85 | 530 |
| Zinc (Zn) | 140 | 140 | 720 |
| Copper (Cu) | 36 | 36 | 190 |

### 2.1. Portable XRF (pXRF)

Spectrometry is a method to detect elements, first established by Goldschmidt [21]. The pXRF instrument was originally invented in the 1970's to locate lead-containing pipes in buildings. Its functionality with multiple elements was further used to map lead in paint [22] and investigating metal contamination in soil and sediments [23–25]. The instrument has further been developed and used for mapping soil and waste contamination [11,26]. Today, one of the main uses of the pXRF instrument is characterization of polluted soils [12,13] and other types of contamination, such as PTE in organic matter, i.e., plants and algae [27,28].

### 2.2. Instrument Description

XRF (X-ray fluorescence) is a non-destructive analytical technique used to determine the elemental composition of materials. The pXRF instrument determines the chemical composition by measuring the fluorescent (or secondary) X-ray emitted from a sample when it is excited by a primary X-ray source. Each of the elements present in a sample produces a set of unique characteristic fluorescent X-rays, like a fingerprint. [29]. The analyses cover the element content from magnesium (Mg, 12) to uranium (U, 92) in the periodic table [29,30]. The detection limit for each element varies, and the limits for the elements in focus here are given in Table 2.

This method for in situ mapping of pollutants in the topsoil of swales was carried out with pXRF, with focus on PTE in soil. Two instruments; Thermo Scienctific$^{TM}$ Niton$^{TM}$ XL3t GOLDD+ XRF Analyzers (#SN67136) and Thermo Scientific Niton XL3t GOLDD XRF Analyzers (#SN36372c) were used to provide comparisons between readings and to improve quality control. The instruments were set for soil analysis and every reading with the instruments were executed for minimum 60 s

for reliable results [30]. The instruments were calibrated with standard reference samples [29] and with Dutch soil standard samples. The latter are samples made from a batch of thoroughly analyzed soil, according to the national background values (Table 1) and following the national regulations for contaminated soil [20].

**Table 2.** Detection limit per element for portable XRF and ICP-MS [29,31,32].

| Metals | Portable XRF ppm (mg/kg) (ThermoFisher) | ICP-MS ppm (mg/kg) (Mineral Laboratories) | ICP-MS ppm (mg/kg) (Mineral Laboratories) |
|---|---|---|---|
| Lead (Pb) | 1 | 0.1 | 0.5 |
| Zinc (Zn) | 4 | 1 | 1 |
| Copper (Cu) | 5 | 0.1 | 0.2 |

### 2.3. In Situ Measurements with Portable XRF

Measurements were performed at a systematic interval of 1 m. To investigate the chemical background concentration of the topsoil, measurements were, for each cross-section, collected on the outside or along the rim of the swale crossing over to the other side, as demonstrated by the measuring tape in Figures 1 and 2. Such measurements will provide the background value and whether any build-up contamination is present in the swale. The profiling was executed as cross-sections from the inlet, the deepest part of the swale and further away from the inlet. The profiling approach is shown in Figure 2 and the method was applied to three locations (Figures 2–4), demonstrating the profiling methodology and results.

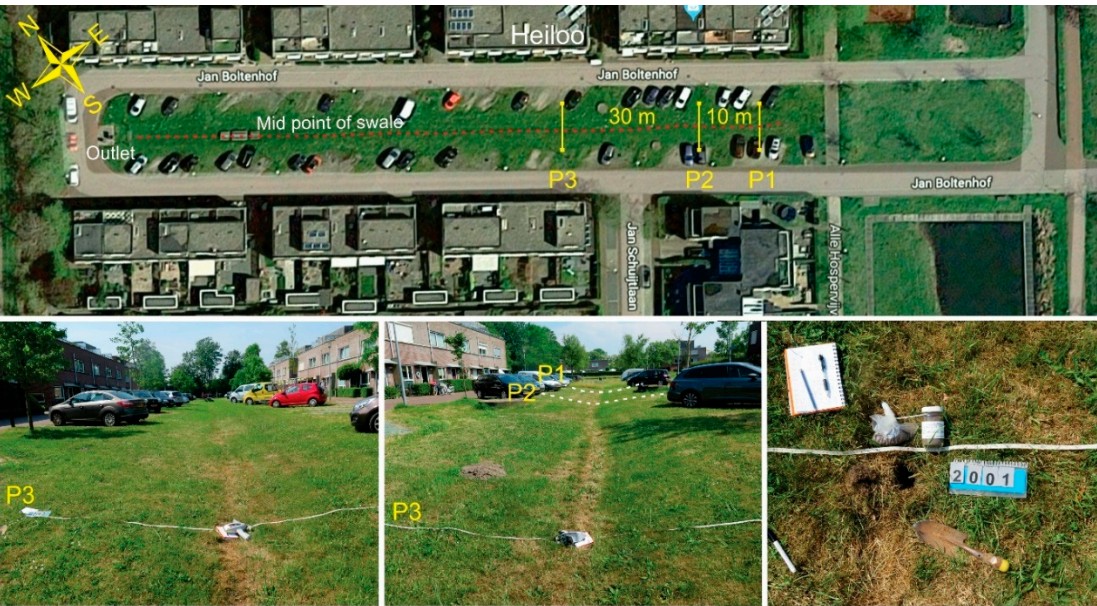

**Figure 2.** The swale at Heiloo is approximately 70 m long and 4 m wide. Three profiles with measurements at each meter were conducted with two different pXRF instruments, XL3#SN67136 and XL3t#SN36372c. Two soil samples for ICP-MS analysis were collected at the same locations at the mid-point of profile 3 for comparison. The soil samples were collected in a Rilsan® bag and a glass jar. A sandblasted stainless-steel spade was used for collecting the soil samples.2.5.2. Limmen

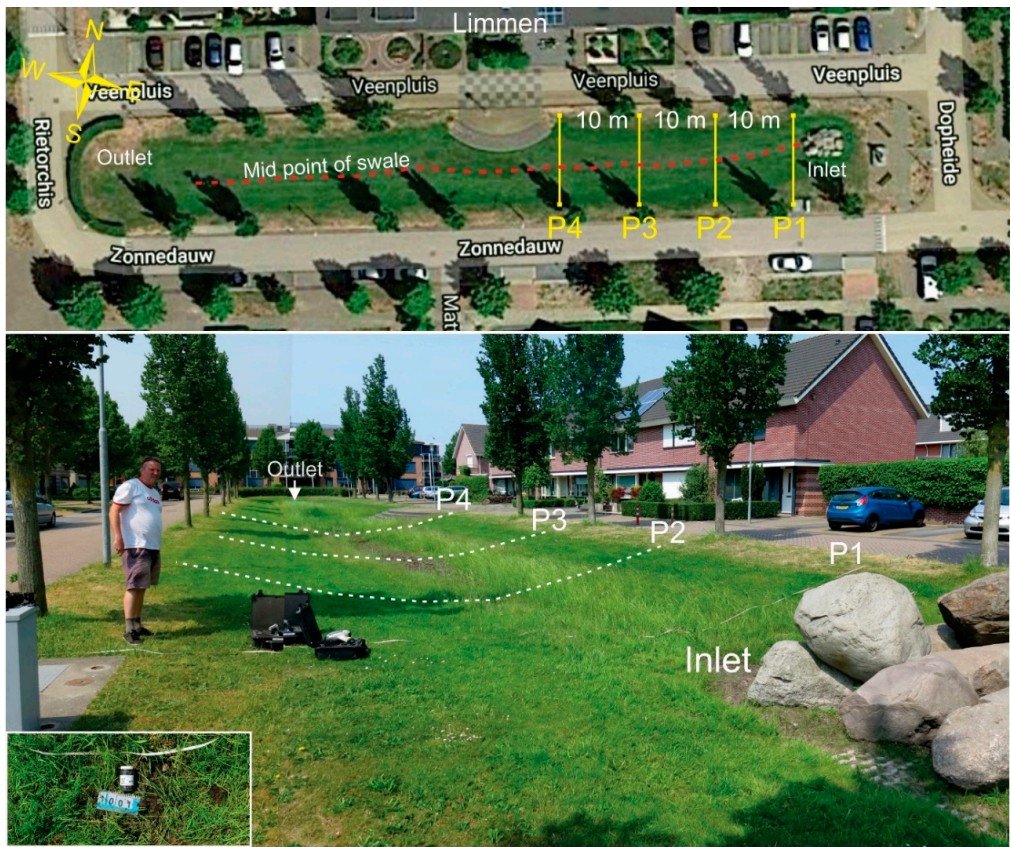

**Figure 3.** The swale at Limmen is approximately 70 m long and 5 m wide. Four profiles with measurements at each meter was collected with two different pXRF instruments. Two soil samples for ICP-MS analysis was collected at mid-point of profile 1.

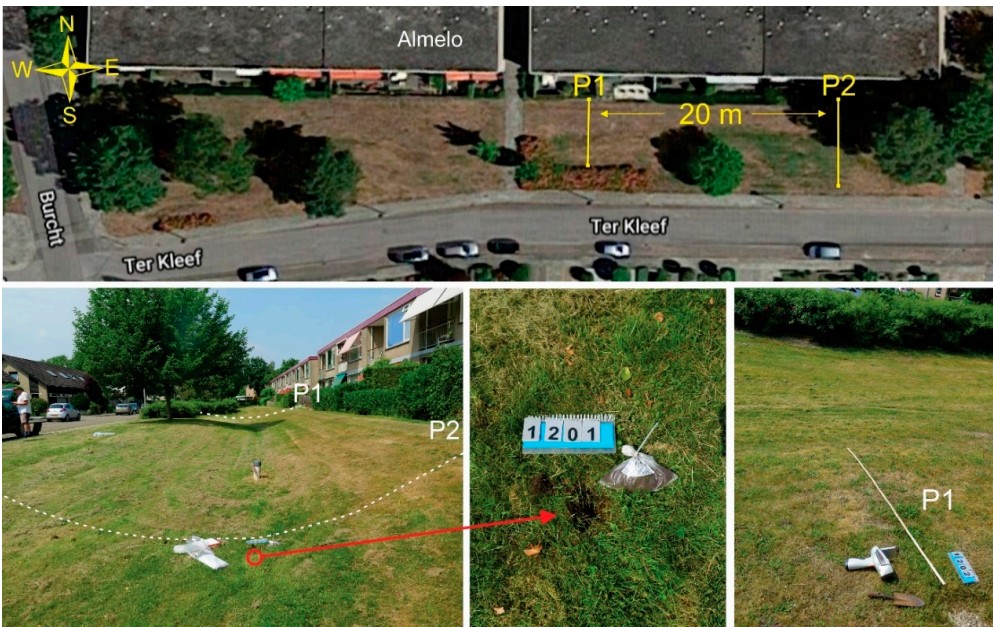

**Figure 4.** The swale at Almelo municipality, is approximately 40 m long and 10 m wide. Two measurement profiles with measurements at each meter were collected with pXRF instruments XL3#SN67136 at P1 and pXRF instruments XL3t#SN36372c at P2. A soil sample for ICP-MS analysis was collected at mid-point of Profile 2.



### *2.4. Soil Sampling for Quality Control*

In order to provide quality control of the in situ pXRF measurements, at least one soil sample was collected at every location to be analyzed in using inductively coupled plasma mass spectrometry (ICP-MS) in a laboratory. The samples were collected either at the inlet or at the deepest point of the swale. The sample collection followed the EuroGeoSurveys Geochemistry Expert Group Sampling protocol [33].

The soil samples of the topsoil (0–3 cm) were collected with a small, sandblasted, stainless steel garden spade and stored in contaminant-free glass container and RILSAN® bags for ICP-MS lab analysis (bottom right photo in Figure 2). The soil samples were dried for 18 days at 26 °C (min 26 °C–max 29 °C), then sieved using a 2 mm plastic sieve before sent for lab analysis. ICP-MS is a type of mass spectrometry which is known for its ability to detect metals with high precision and low detection limits (Table 2). The soil samples with metals were dissolved in acid (Aqua Regia Digestion). Then, the digestion liquid was analyzed with an ICP-MS instrument, as described by Flem et al. [34]. In the lab, the soil samples were then each measured with the pXRF (XL3#SN67136) five times. The sample material was blended before each measurement to determine the heterogeneity of the sample material.

### *2.5. Data Collection at Locations*

The three locations from the study are described here, providing examples for the method. Detailed characteristics of each swale studied are given in Table 3.

**Table 3.** Detailed characteristics for the swales studied.

| Location (Municipality/Street) | Heiloo Jan Boltenhof | Limmen Zonnendauw | Almelo Ter Kleef |
|---|---|---|---|
| Areal use | Residential | Residential | Residential |
| Year of construction | 1999 | Approximately 2000 | 2002 |
| Estimated storage [volume m$^3$/connected area m$^2$] | 20 mm | 15 mm | 10 mm |
| Type of infiltration test | Observation * | Observation * | Long term monitoring (loggers) |
| Infiltration rate (m/day) | >0.3 m/day * | >0.3 m/day * | 0.15–0.49 m/day |
| Soil type | Sandy soil | Clay/organic soil | Sandy soil |
| More information | https://climatescan.org/projects/135/detail | https://climatescan.org/projects/3/detail | https://climatescan.org/projects/941/detail |

* Observation/interview by municipality.

### 2.5.1. Heiloo

The swale at Heiloo municipality is located in a residential area in the north-western part of the Netherlands (https://climatescan.org/projects/135/detail). This swale is a grassy field, approximately 70 m long and 4 m wide with car parking along each side (Figure 2). The surface water inlet is not from one point, but all along both sides. The outlet is located at the north-western point of the swale. Measurements from this swale were collected with two pXRF instruments (#SN67136 & #SN36372c) used at each of the three profiles for comparable results. Profile 1 (P1) is close to the south-eastern end of the swale, profile 2 (P2) and profile 3 (P3) are 10 m and 40 m from P1 (Figure 2). Duplicate measurements were taken, first on top of vegetation and then in direct contact with topsoil after the removal of vegetation. This was conducted to determine the influence of vegetation on the results. Further, it was decided to continue measurements only on topsoil because results from in situ topsoil measurements could be correlated to laboratory analysis of soil samples.

The swale at Limmen is located in a residential area in the northwestern part of the Netherlands (https://climatescan.org/projects/3/detail). It is a large swale that measures approximately 70 m long

and 5 m wide (Figure 3). The swale is surrounded by local roads and provides aesthetic effect by creating a park-like setting with grass vegetation. The swale has a clear outer rim, where the grassy field is kept short while the central part of the swale has taller vegetation (Figure 3). The part with tall vegetation is commonly flooded by surplus stormwater, and the deepest part is without vegetation due to frequent water cover. This swale has a one-point inlet at the eastern side and a one-point outlet at the western side. The measurements at the swale were conducted along four profiles with an interval of 10 m between the profiles. The pXRF measurements were conducted from left (south) to right (north) as shown in Figure 3. Profile 1 was measured with two instruments (#SN67136 & #SN36372c, Figure 3) to enable comparison of the different instruments.

### 2.5.2. Almelo

The swale at Almelo municipality is located in a residential area in the eastern part of the Netherlands (https://climatescan.org/projects/941/detail). This swale is a grassy field, approximately 40 m long and 10 m wide with residential homes along the northern side and a street at the southern side (Figure 4). The surface water inlet is located at two points, connected to the gutter from a house on the northern side and located at the starting point (0 m) of each profile. The outlet is at the western point of the swale (Figure 4). Two measurement profiles, located 20 m from each other, were conducted with two pXRF instruments, #SN67136 at Profile 1 and instruments #SN36372c at Profile 2.

## 3. Results and Discussion

### 3.1. Results Heiloo

At the Heiloo location three parallel profiles were measured at a 0, 10, and 30 m distance (Figures 2 and 5A–C) where all profiles were measured by the two pXRF instruments (#SN67136 & #SN36372c). To test out the influence of vegetation, all profiles at location Heiloo were executed with measurements on top of vegetation and directly on soil with vegetation removed, scraped off or just below roots.

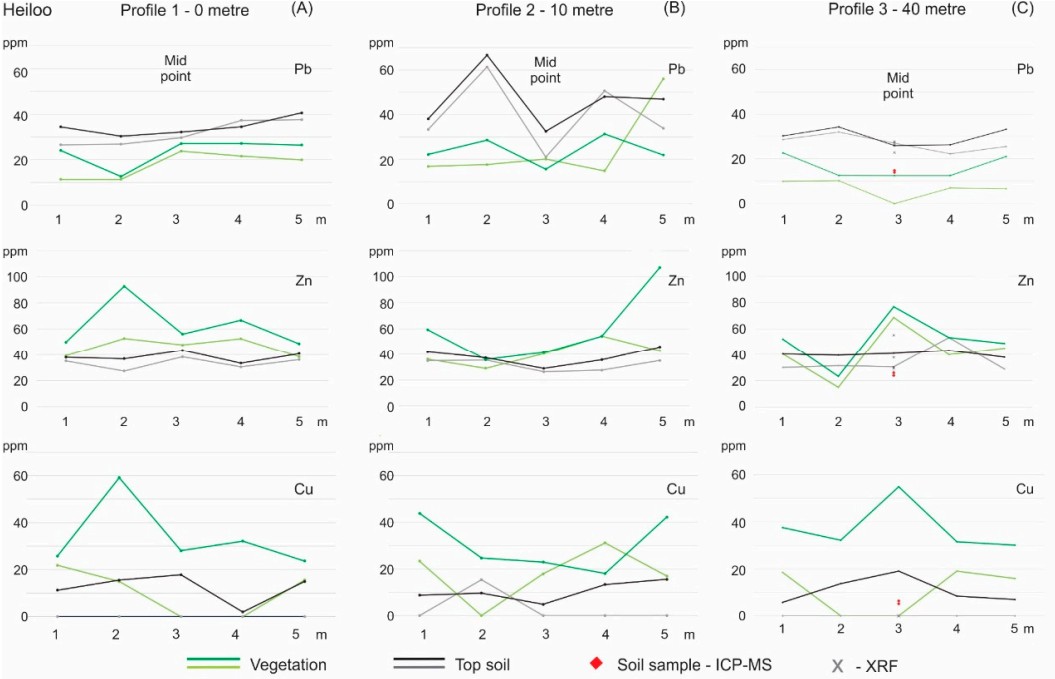

**Figure 5.** The profiles at location Heiloo were measured with two different pXRF instruments, on top of vegetation (green lines) and on topsoil (black and grey lines) where vegetation is removed. Elements in graphs from top to bottom: Pb—Lead, Zn—Zinc, and Cu—copper. See Figure 2 for location of profiles.

(**A**). Results from profile 1, closest to the inlet (**B**). Results from profile 2, 10 m from profile 1. (**C**). Results from profile 3, 40 m from profile 1. Two soil samples were collected at midpoint of swale (deepest point) for quality control of method. Results from the soil samples are marked with red dots for ICP-MS analysis and X for the repetitive pXRF measurements. For copper (Cu) several measurements are below detection of 5 ppm (Table 2). All results are below national threshold values for topsoil (Table 1).

The results from all profiles shown in Figure 5A–C indicate that lead (Pb) concentrations are lower in the vegetation than on topsoil. For Zinc (Zn) and Copper (Cu) the results are not so consistent. This is coherent with the findings of lead in vegetation by Zimdahl [35]. Other elements such as sulphur (S), calcium (Ca), potassium (K), chromium (Cr) show the opposite relation and are prone to be elevated in vegetation [16,35]. PTE appear to be accumulated in the topsoil, which is the media that people are exposed to [2,5,8–10]. Therefore, the uppermost topsoil (0–3 cm) is of interest, and no more measurements on vegetation were conducted.

The results from Heiloo shown in Figure 5A–C do not reveal a consistent pattern. As shown for profile 3, the concentration of Pb from the laboratory analyses on the soil sample (ICP-MS, red dots in Figure 5C) is lower than that in the pXRF in the field but matches the result from the pXRF measured in the lab (marked with X in Figure 5C). The lack of consistency of the pattern, as is found in the two other locations Limmen and Almelo, is most likely due to the lack of clear water inlet and little water being discharged into the swale. There were no signs of the swale being frequently flooded since the swale was evenly vegetated (Figure 2). All values measured for lead, zinc, and copper are below the national intervention value (Table 1).

*3.2. Results Limmen*

At the Limmen location, four parallel profiles with 10 m intervals were measured (Figures 3 and 6A–C). The profiles were measured from one side of the swale to the other, and the results show that for lead and zinc values are low on both outside rims of the swale and increase in the middle of the swale (Figure 6A,B). Values are high near the inlet and at the deepest point of the swale, in profile 1 and 2 (Figure 6A,B). For lead and zinc, there is a build-up in the middle part, where the swale is most frequently flooded. This does not apply to copper (Cu), which has no consistent pattern as shown in Figure 6C. The values from the soil sample by lab ICP-MS are higher than in situ pXRF measurements (red dots in Figure 6A–C). The lab measurements on the soil sample by the pXRF (XL3#SN67136) are on the same level as the lab analysis, as shown in Figure 6A–C, marked with X. Values of lead and copper are close to the national background values, but all values measured at Limmen are below intervention levels (Table 1).

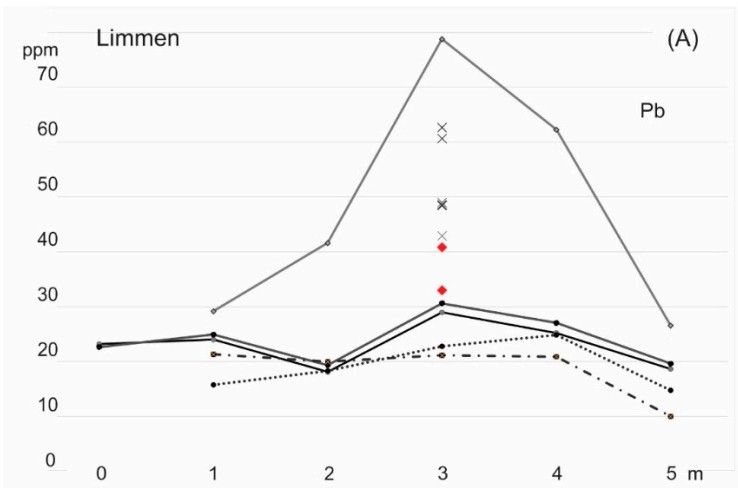

**Figure 6.** *Cont.*

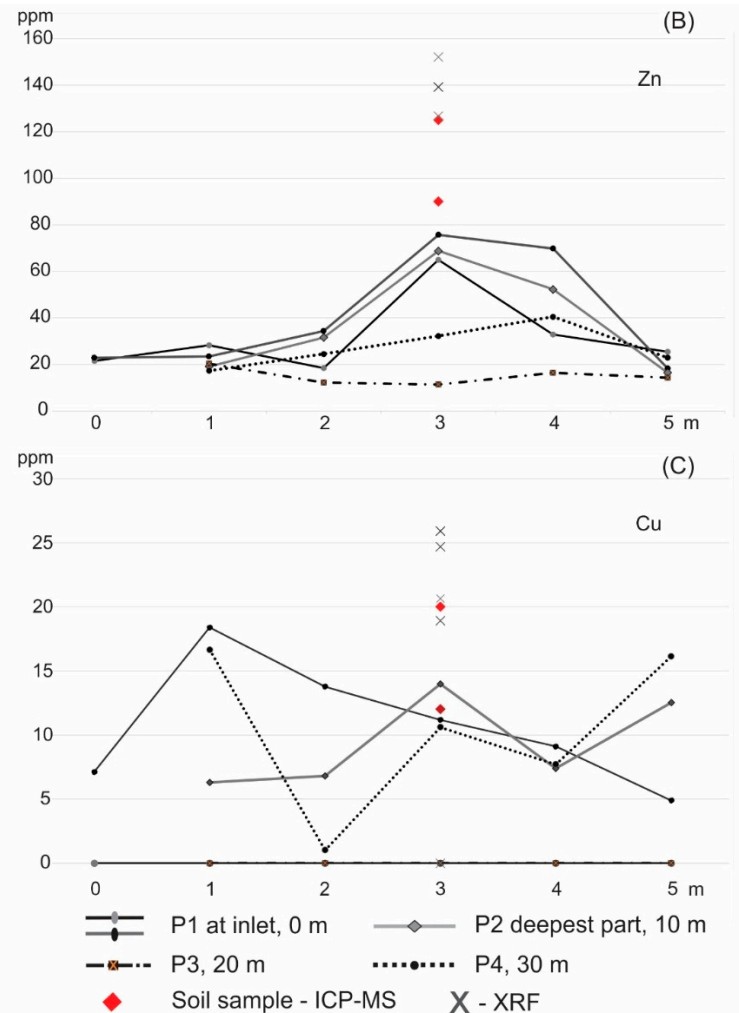

**Figure 6.** Results for lead, zinc and copper from location Limmen. Four profiles were measured on topsoil, where profile 1 was measured with two different pXRF instruments for comparison. P1 is a cross-section covering the inlet at 0 m, P2 cross-cut the deepest part of the swale at 10 m, where P3 and P4 are located at 20 and 30 m, respectively. The location of the profiles is shown in Figure 3. Two soil samples were collected at the midpoint of the swale in profile 1, close to the inlet for quality control of method. Results from the soil samples are marked with red dots for ICP-MS analysis and X for the repetitive pXRF measurements. (**A**). Results for lead (Pb). (**B**). Results for zinc (Zn). (**C**). Results for copper (Cu). For copper (Cu) several measurements are below the detection limit of 5 ppm (Table 2), expressed as zero in this figure. National intervention values for topsoil are 530 ppm, 720 ppm and 190 ppm for Pb, Zn and Cu, respectively (Table 1).

### 3.3. Results Almelo

At the Almelo location, two parallel profiles separated by 20 m were measured (Figures 4 and 7). This swale is different from those in Heiloo and Limmen, which have a trench shape, where the side facing the residential homes slopes gradually towards the mid-point, while the opposite side extends flat towards the road (Figure 4). The inlet points are positioned at the end of numerous residential gutters, which are the starting point of profile 1 and 2. This is reflected in the results, Figure 7, where the highest metal concentrations in the soil are close to the inlet and decrease further away from the inlet. Copper (Cu) differs from this trend, as the highest concentration is located at the mid-point (Figure 7). The values from the ICP-MS analysis of the soil sample (taken from profile 2) resemble the result from the pXRF. The values measured for lead, zinc, and copper in this swale are close to or higher than the national background value, but below the intervention values (Table 1).

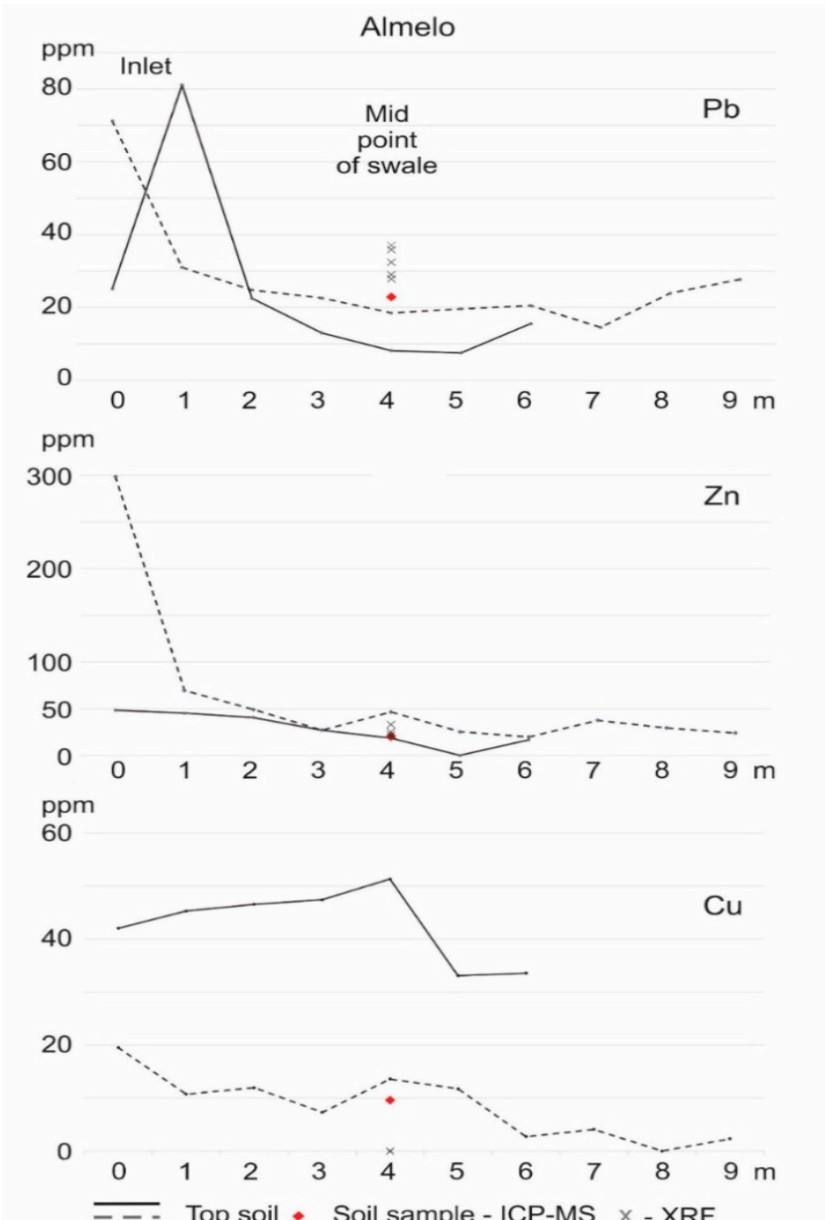

**Figure 7.** Results from Almelo. Both profiles start at an inlet (0 m). Profile 1. Profile 1 and 2 are situated approximately 20 m apart. The soil sample, marked with red dots analyzed by ICP-MS and X for repetitive readings in lab with pXRF instrument, was collected at mid-point of profile 2. Elements in graphs from top to bottom: Pb—Lead, Zn—Zinc and Cu—copper. Results show values above national target value, but well below intervention values (Table 1).

*3.4. Discussion*

The result shows that the highest concentration of pollutants is close to the inlet(s), based on the pXRF measurements, which is consistent with the findings of Jones and Davis [10] and Tedoldi et al. [9]. Jones and Davis [10] and Tedoldi et al. [9] also observe that the spatial distribution of pollutants varies over a short distance and can vary greatly within SuDS. The variation of spatial distribution of pollutants in swales is confirmed, where great variations over short distances (1 m) can be observed. The distribution of the pollutants is controlled by the pathways of the water in the swale, with highest measured values close to the inlet and at the deepest point of the swale, where the occurrence of surplus water is most frequent and has the longest duration. Since the water flow path controls the distribution of the pollutants it is important to locate the profiles in such manner that it covers the inlet

and the deepest part of the swale, as exemplified in Figures 2–4. These results confirm that stormwater is a significant contributor of pollutants to SuDS, as shown by Boogaard et al. [5].

This study has focused on the occurrence of PTE in the topsoil layer and should be followed up by sampling at greater soil depths and vertical profiling, as demonstrated by Ingvertsen et al. [36], Jones and Davis [10] and Tedoldi et al. [9]. These studies show that PTE are trapped in the uppermost layers of the soil, and the concentration decreases with depth in the uppermost 0–30 cm, with the highest concentration in topsoil [9,10,36], as PTE in stormwater are prone to be particle-bound [5].

Soil samples were collected and analyzed according to Demetriades and Birke [33], which is a common standard procedure for geochemical mapping as shown by Flem et al. [34]. The variations in the different elements, especially copper (Cu), demonstrate the importance of control samples of soil with laboratory analysis, such as ICP-MS. Work by Radu and Diamond [23], Bull, Brown and Turner [37], Turner and Solman [38], Rincheval, Cohen and Hemmings [27], Turner et al. [28,39] and Lemière et al. [40] have documented the pXRF tool's reliability. Researchers analyzed many forms of media, including algae, bark, soil and sediments, and all demonstrate that the results produced by the pXRF were as reliable as those produced in laboratory, using methods, such as ICP-MS. Based on the comparative analyses presented in [26–28,37–40], we argue that in situ measurements with the pXRF are valid as a quick-scan method for mapping PTE pollutants in SuDS. Radu and Diamond [26] have conducted an extensive study correlating pXRF results with results from laboratory atomic absorption spectrophotometry (AAS), giving pXRF credential as a predictable in situ measuring tool. Lemière et al. [40] documents well that high water content influences the results of pXRF. Their work shows the correlation of water content, pXRF measurements and laboratory analysis [40]. Taken the water content into consideration, Lemière et al. [40] still conclude that pXRF is an efficient and reliable tool for in situ mapping of contaminated sediments.

This work is part of a national study [17] where two national workshops on this research-based study contributed to the re-evaluation of the design or to new guidelines for design, management and maintenance of sustainable urban drainage systems, to improve the water quality of the receiving water bodies [41,42]. Portable XRF measurements provide a rapid in situ analysis of the concentrations of a range of elements, which can help lead to a rapid response to a hazardous waste situation. This notion has also been pointed out by Kalnicky and Singhvi [11]. The traditional soil samples and lab analysis are time-consuming as well as costly compared to in situ pXRF measurements. Therefore, this suggested method using pXRF provides a foundation to limit area for follow-up investigations including in-depth profiling, as demonstrated by Tedoldi et al. [8,9], Jones and Davis [10] and Ingvertsen et al. [36]. The values of the detected PTE described here are well below the Dutch threshold values for intervention for lead, zinc and copper (Table 1). Hence, the pXRF quick scan analytical procedure could be used for water management and water quality control to prevent build-up of soil pollution entering the groundwater body to comply with the Water Framework Directive [41] and the CIRIA-The SuDS manual [2].

## 4. Conclusions

A methodology of in situ measurement of Potential Toxic Elements (PTE) pollutants in Sustainable urban Drainage Systems (SuDS) using portable XRF (pXRF) instruments at three pilot locations in the Netherlands is presented. This is a cost and time-efficient method that gives immediate results, as pointed out by Bernick et al. already in 1995 [24]. The pXRF instrument's detection limits are well below national threshold values which makes this method suitable for the purpose of measuring PTE content in soils in SuDS for management purposes. With this quick-scan method, traditional soil sampling and laboratory analyses can be minimized to control samples. This thus suggests a method for a thorough chemical characterization of PTE of topsoil in SuDS, exemplified by swales.

Three swales have been investigated with the pXRF method. The results confirm a substantial variation of spatial distribution of PTE pollutants; the storm-water inlet and water volume control the distribution in swales. When in the field, the measurement profiles should be adjusted according

to the design of the SuDS, making sure that the profiles cover the inlet—the deepest section as well as the outer rim—to represent the highest and lowest possible values of built up of PTE pollutants. The measurements along the profiles should be executed systematically with a set interval. Control samples of soil should be collected and analyzed in the laboratory to validate the pXRF quick scan.

This quick scan pXRF mapping methodology of topsoil will indicate if the topsoil is polluted and whether the concentrations exceed national or international standards. If pollutant values are found above threshold values, a follow-up investigation with more detailed mapping, both with quick scan, soil samples on the surface and in depth, is recommended before clean-up is initiated.

The research results from this applied methodological approach will help national water authorities to control and improve the water quality in accordance with the EU Water Framework Directive [41]. In addition, the results will help improve national and international guidelines for the design, construction and maintenance of SuDS, the importance of which has been recognized by several stakeholders from water authorities and municipalities attending two recent national workshops [18].

**Author Contributions:** Both authors have equally participated in the preparation of the study and execution of fieldwork with data collection. Data have been analyzed together and results discussed. G.V.; writing—original draft preparation, and preparing most figures, with continuous contribution from F.C.B. The manuscript has been revised by both authors. All authors have read and agreed to the published version of the manuscript.

**Funding:** This research is funded by the Joint Programming Initiative Water Challenges for a changing world: WaterWorks 2014, INXCES—INnovations for eXtreme Climatic EventS, Grant JPI Water—Water Works ERA-NET Cofund 2014; Norwegian Research Council No. 258647, Nederlandse Organisatie voor Wetenschappelijk Onderzoek No. ALWWW.2014.2. National workshops, publications and meetings with several stakeholders are funded by WaterCoG) WaterCo-Governance) co-funded by the North Sea Region Programme 2014–2020.

**Acknowledgments:** We thank Hanze University of Applied Sciences Groningen and the Geological Survey of Norway and Tauw for support for this work. The authors would like to thank the involved municipalities in this study and the Dutch national organisations Stichting Toegepast Onderzoeks Waterbeheer (STOWA) and Stichting RIONED for co-organizing the national workshops. Great appreciation to M.A. at the Geological Survey of Norway (NGU) for constructive feedback on this paper, and to A.L.-D. (NGU) for improving this paper.

**Conflicts of Interest:** The authors declare no conflict of interest. The funders had no role in the design of the study; in the collection, analyses, or interpretation of data; in the writing of the manuscript, or in the decision to publish the results.

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
