# Peer review of "Portable XRF Quick-Scan Mapping for Potential Toxic Elements Pollutants in Sustainable Urban Drainage Systems: A Methodological Approach"

_sci, doi:10.3390/sci2030064_

Round 1

Reviewer 1 Report

General comments:

In a context where sustainable urban drainage system are more and more implemented, to filtrate and infiltrate storm water, there is a need to control the charge of contaminants in such system in order to make sure that they do not allow contaminants to impact groundwater. In this frame, the paper proposes a methodology to map heavy metals in sustainable urban drainage system (SuDS), based on in situ measurement on topsoils with a portable XRF tool.

The methodological approach is rather well described, but it should be reorganized to allow a better understanding. The results are rather disappointing as no maps are presented. The section is sometimes difficult to follow: different pXRF tool used (difference), comparability with lab analysis not clear, applicability. It should be reorganized and completed. The beginning of conclusion looks like an abstract. The authors should focus on more general conclusions: interest of proposed methodology, recommendations.

In general, the authors refers very much to “this” study”, suggesting a report extract.

Detailed comments:

  • Title:

The title sounds in accordance with the objectives of the paper.

Should nevertheless the abbreviation SuDS be used?

Suggestion: “Quick in situ trace elements mapping for urban drainage system management: a methodological approach”

Instead of heavy metal, I prefer the term trace elements (more general).

  • Introduction:

It should be revised to end with the objectives of the paper.

1st and 2nd paragraphs writing ok.

The interest of checking the trace elements content of topsoils in SuDS for management purposes should be explained before giving the objectives of the paper (I would suggest putting it in between 1st and 2nd paragraphs).

The figure doesn’t appear necessary in the introduction part. It could be placed in the Material and method section to present the functioning of the swales and explain the sampling strategy.

Avoid repetition (eg pXRF quick and cost efficient).

The authors could add more references on pXRF use, like:

  • Lemière B. (2018), Bruno Lemiere. A Review of pXRF (Field Portable X-ray Fluorescence) Applications for Applied Geochemistry. Journal of Geochemical Exploration, Elsevier, DOI: 10.1016/j.gexplo.2018.02.006
  • RAMSEY M.H., & BOON K.A., 2012 - Can in situ geochemical measurements be more fit-for-purpose than those made ex situ? Applied Geochemistry 27, p. 969–976.
  • JEAN-SORO L., C. LE GUERN, B. BECHET, T. LEBEAU, M.F. RINGEARD, 2014 - Origin of trace elements in an urban garden in Nantes, France, Journal of Soils and Sediments, DOI 10.1007/s11368-014-0952-y.

  • Materials and methods

The 2nd paragraph of the introduction part of the section should be moved to the general introduction (as context), or to the discussion, or to the conclusion.

The section should be reorganized. A suggestion of organization is given here after

  1. Principle of sampling strategy
    1. Functioning of swales
    2. General sampling strategy (2.3.)
  2. Quantification of trace elements
    1. In situ measurements
      1. Principle of pXRF (agragetion of 2.1. and beginning of 2.2)
      2. Tools used

Explain the difference between the tools

  • Analysed samples

Preparation, With/without vegetation

  1. Lab analysis
    1. Preparation, ICP MS…
  2. Study sites
    1. Description
    2. Sampling/measurements

Precise here period of measurements => soils dry or wet, organic matter…

  1. Interpretation method

Comparison between in situ measurements and lab analysis

Comparison to threshold values

Mapping ?

  • Results

It is very disappointing to see no map of trace elements for each site, whereas the title and objectives announce it.

Although other studies have demonstrated the comparability of pXRF measurements and ICP/MS analyses, some matrix effect might disturb the correlation (influence of organic matter content, humidity of soil…, see eg. Lemiere et al, 2014). The proposed figure do not make it easy to get convinced that the correlation is sufficient. How about a figure compiling the correlation between in situ measurements and lab analysis?

Lemiere, B., Laperche, V., Haouche, L. & Auger, P., 2014 - Portable XRF and wet materials: application to dredged contaminated sediments from waterways. Geochemistry: Exploration, Environment, Analysis 14, 257-264. 

It seems that the tests carried out in each site have given purposes=> reorganize the section according to these purposes

  1. Comparability of pXRF and ICP/MS measurements – reliability of pXRF in situ measurements
  2. Influence of pXRF tool and support of measurement (vegetation/topsoil)

Test carried out on Heiloo site.

Conclude on these aspects

  1. Position of measurement profiles regarding inlet, outlet and other site specificities
  2. Discussion
    1. It should focus on the methodological aspects

At the end of the discussion, the authors write: “The XRF quick scan analytical procedure could be used for water management and water quality control to comply with the Water Framework Directive [16] and the CIRIA‐The SuDS manual [2].” : one may understand that you can analyze water with pXRF. The sentence should be modified to avoid the possibility of such interpretation.

  • Conclusion

The beginning of conclusion looks like an abstract. The authors should focus on more general conclusions: interest of proposed methodology, recommendations for current sampling, as well as future sampling (survey).

Adding remarks: How to ensure an efficient survey? Soil sampling conduct to soil removal. How to make sure that sampling nearby in the future (survey) will provide comparable results => need to verify short distance spatial variations of contents. Have you made such tests in situ?

  • Figures and tables:

Current figures difficult to read. The Figures should be revised to fit with the revised organization of results and discussion section.

Fig 5: Size much too large (should fit in 1 p max). rem : Size of fig 7 is convenient.

Separate vegetation/topsoil results (support and in situ tool) from pXRF/ICP-MS results (measurement/analytical tool) to make results easier to understand. Dotted lines for vegetation will make reading easier too.

Cu in profile 1, 2 and 3 are set to 0. This must be changed to detection limit.

Fig 6: too many lines, difficult to read. Same comments as for fig 5 (pXRF/ICP-MS comparison). Fig. too large too.

Fig 7: what do the both profiles correspond to?

How about indicating threshold values in Fig 5 to 7?

Add at least maps comparing results to Dutch threshold values could be shown in addition (green dots below threshold value, yellow, red above intervention value.

Comparison between pXRF and ICP/MS results should be shown.

  • References:

Well documented paper. Some suggestions nevertheless (see comments in the introduction and discussion sections)

Author Response

Sorry, added same responce to reviewer in both boxes. This is responce to reviewer 2: Thank you for review of the paper. Yes, we agree on all your comments, and will take this into consideration when the results of the national swale mapping is prresented. Other authors have studied several of the issues you point out, and referred to in this paper. Thank you very much for your review of this paper. pXRF is a well known instrument, and Method, within the Field of geochemistry, but unknown in other. This work focuses on establishing a mapping Method of potential toxic elements aimed at those working within water management and SuDS. We agree on Your comments, but have chosen to describe the Method and will present the results seperately in additional work. Thank you for improving the paper and adding valuable References. Specific feedback on comment are listed below. Comments and Suggestions for Authors General comments: In a context where sustainable urban drainage system are more and more implemented, to filtrate and infiltrate storm water, there is a need to control the charge of contaminants in such system in order to make sure that they do not allow contaminants to impact groundwater. In this frame, the paper proposes a methodology to map heavy metals in sustainable urban drainage system (SuDS), based on in situ measurement on topsoils with a portable XRF tool. The methodological approach is rather well described, but it should be reorganized to allow a better understanding. The results are rather disappointing as no maps are presented. Answer: We have chosen to only address the method in this paper. The results will be published in separate papers. This is to not make the papers too lengthy. I agree on the suggestion showing results in maps. However, this will be done in a follow-up paper. The section is sometimes difficult to follow: different pXRF tool used (difference), comparability with lab analysis not clear, applicability. It should be reorganized and completed. The beginning of conclusion looks like an abstract. – conclusion is altered after suggestion. The authors should focus on more general conclusions: interest of proposed methodology, recommendations. In general, the authors refers very much to “this” study”, suggesting a report extract. Answer: Point taken. Changed accordingly. Detailed comments: • Title: The title sounds in accordance with the objectives of the paper. Should nevertheless the abbreviation SuDS be used? Suggestion: “Quick in situ trace elements mapping for urban drainage system management: a methodological approach” Answer: since this method can be used for several SuDS we would like to keep the title Instead of heavy metal, I prefer the term trace elements (more general). Answer: changed to Potential Toxic Elements (PTE) • Introduction: It should be revised to end with the objectives of the paper. 1st and 2nd paragraphs writing ok. The interest of checking the trace elements content of topsoils in SuDS for management purposes should be explained before giving the objectives of the paper (I would suggest putting it in between 1st and 2nd paragraphs). The figure doesn’t appear necessary in the introduction part. It could be placed in the Material and method section to present the functioning of the swales and explain the sampling strategy. Avoid repetition (eg pXRF quick and cost efficient). Answer: Changes as suggested The authors could add more references on pXRF use, like: • Lemière B. (2018), Bruno Lemiere. A Review of pXRF (Field Portable X-ray Fluorescence) Applications for Applied Geochemistry. Journal of Geochemical Exploration, Elsevier, DOI: 10.1016/j.gexplo.2018.02.006 • RAMSEY M.H., & BOON K.A., 2012 - Can in situ geochemical measurements be more fit-for-purpose than those made ex situ? Applied Geochemistry 27, p. 969–976. • JEAN-SORO L., C. LE GUERN, B. BECHET, T. LEBEAU, M.F. RINGEARD, 2014 - Origin of trace elements in an urban garden in Nantes, France, Journal of Soils and Sediments, DOI 10.1007/s11368-014-0952-y. Answer: we would like to thank the reviewer for these very relevant references, they are added to the text. • Materials and methods The 2nd paragraph of the introduction part of the section should be moved to the general introduction (as context), or to the discussion, or to the conclusion. Thank you for this suggestion. This work focus on the method, which is established within some disciplines but new in water management and SuDS. Therefore, the amount of results presented here is limited. After review of several reviewers, we choose to keep the present structure of the paper, but will keep your suggestion in mind in further work with the results. The section should be reorganized. A suggestion of organization is given here after 1. Principle of sampling strategy 1. Functioning of swales 2. General sampling strategy (2.3.) 2. Quantification of trace elements 1. In situ measurements 2. Principle of pXRF (agragetion of 2.1. and beginning of 2.2) 3. Tools used Explain the difference between the tools • Analysed samples Preparation, With/without vegetation 1. Lab analysis 1. Preparation, ICP MS… 2. Study sites 1. Description 2. Sampling/measurements Precise here period of measurements => soils dry or wet, organic matter… 4. Interpretation method Comparison between in situ measurements and lab analysis Comparison to threshold values Mapping ? • Results It is very disappointing to see no map of trace elements for each site, whereas the title and objectives announce it. Answer: We have deliberately not included maps of results in this paper to focus on the method for mapping and not the results of the mapping. Even though pXRF is an established tool within geochemistry, it is a new tool within SuDS and water management for mapping of PTE. Professions for SuDS and water management is our aimed audience to improve their approach of maintenance. Although other studies have demonstrated the comparability of pXRF measurements and ICP/MS analyses, some matrix effect might disturb the correlation (influence of organic matter content, humidity of soil…, see eg. Lemiere et al, 2014). Answer: Yes, we agree and are aware of the matrix disturbance. However, for the purpose of the method the consideration are the national threshold values. These thresholds are far above disturbance limits. Will include in discussion to stronger point out the possible factors of disturbance of the results. The proposed figure do not make it easy to get convinced that the correlation is sufficient. How about a figure compiling the correlation between in situ measurements and lab analysis? Answer: Good point. This will be included in the results paper. For the data presented here, the number of data are not large enough to get a significant correlation. Previous reviewers have commented this as well. Lemiere, B., Laperche, V., Haouche, L. & Auger, P., 2014 - Portable XRF and wet materials: application to dredged contaminated sediments from waterways. Geochemistry: Exploration, Environment, Analysis 14, 257-264. Thank you for a good reference. It seems that the tests carried out in each site have given purposes=> reorganize the section according to these purposes 1. Comparability of pXRF and ICP/MS measurements – reliability of pXRF in situ measurements 2. Influence of pXRF tool and support of measurement (vegetation/topsoil) Test carried out on Heiloo site. Conclude on these aspects 3. Position of measurement profiles regarding inlet, outlet and other site specificities 4. Discussion 1. It should focus on the methodological aspects At the end of the discussion, the authors write: “The XRF quick scan analytical procedure could be used for water management and water quality control to comply with the Water Framework Directive [16] and the CIRIA‐The SuDS manual [2].” : one may understand that you can analyze water with pXRF. The sentence should be modified to avoid the possibility of such interpretation. Answer: Thank you for this feedback. Sentence is changed to clear misunderstanding. • Conclusion The beginning of conclusion looks like an abstract. The authors should focus on more general conclusions: interest of proposed methodology, recommendations for current sampling, as well as future sampling (survey). Answer: We have changes, according to suggestions. Adding remarks: How to ensure an efficient survey? Soil sampling conduct to soil removal. How to make sure that sampling nearby in the future (survey) will provide comparable results => need to verify short distance spatial variations of contents. Have you made such tests in situ? Answer: This will be included in follow up work that will focus on the results. • Figures and tables: Current figures difficult to read. The Figures should be revised to fit with the revised organization of results and discussion section. Fig 5: Size much too large (should fit in 1 p max). rem : Answer: Size of figure 5 is changed to fit one page. Size of fig 7 is convenient. Separate vegetation/topsoil results (support and in situ tool) from pXRF/ICP-MS results (measurement/analytical tool) to make results easier to understand. Dotted lines for vegetation will make reading easier too. Cu in profile 1, 2 and 3 are set to 0. This must be changed to detection limit. Answer: Cu below LOD commented in figure text Fig 6: too many lines, difficult to read. Same comments as for fig 5 (pXRF/ICP-MS comparison). Fig. too large too. Fig 7: what do the both profiles correspond to? Answer: The aim of the paper is to demonstrate the method for pXRF to map potential toxic elements in SuDS. The results are primarily shown to demonstrate the spatial distribution of the elements, and that the water flow controls the pattern of build-up. How about indicating threshold values in Fig 5 to 7? Answer: All results are well below intervention values, thereby no point in adding in the figures. Add at least maps comparing results to Dutch threshold values could be shown in addition (green dots below threshold value, yellow, red above intervention value. Comparison between pXRF and ICP/MS results should be shown. Answer: We have chosen to only address the method in this paper. The results will be published in separate papers. This is to not make the papers too lengthy. I agree on the suggestion showing results in maps. However, this will be done in a follow-up paper. • References: Well documented paper. Some suggestions nevertheless (see comments in the introduction and discussion sections) Answer to reviewer: Thank you for your feedback, that contributes to improve the paper. The title points out that As pointe

Reviewer 2 Report

This paper is quite interesting because the authors propose a way to easily measure in situ pollutants. They use portable XRF equipment and validate the measurements with ICP-MS in the laboratory. This approach facilitates the monitoring of the heavy metals Pb, Zn and Cu in urban contexts.

The text is well-structured, clear and easy to read. They make a good sampling and they calibrate their XRF equipment with different Dutch soil standard samples.

The only aspect I could point to has to do with the precision and reproducibility of the portable XRF instrument’s measurements. The authors indicate that the detection limits of their equipment are 1, 4 and 5 ppm for Pb, Zn and Cu respectively, as the companies indicate. For ideal samples these values would be accurate, but in complex samples such as soils they could increase. It would be good to measure some soil standard samples in the laboratory to check this limit, although this is not critical, because the recorded values are generally above those limits.

A more important aspect would be to check the reproducibility and the accuracy of the equipment. The portable XRF instrument has a narrow spot and the soil samples are usually very heterogeneous which could imply a poor representation of the results. We suggest making a few measurements of some soil standards and present the certified values and the recovered values in a table to generate reassurance in this aspect.

The authors’ idea of a validation of field measurements in comparison with the ICP-MS lab measurements is a good proposal. In spite of this, sometimes validation could be inaccurate in other soils, due to a heterogeneous structure, aggregates, varying porosities and diverse amounts of organic matter, which makes reproducibility more difficult. This validation is satisfactorily checked by the authors in this study, but it should maybe be checked again whenever soils of different mineralogy, granulometry or structure were measured. This aspect could be reflected in the text.

Another interesting improvement for future works would be including spatial measurements to obtain a distribution of the pollutants in swales.

Author Response

xx Thank you very much for your review of this paper. pXRF is a well known instrument, and Method, within the Field of geochemistry, but unknown in other. This work focuses on establishing a mapping Method of potential toxic elements aimed at those working within water management and SuDS. We agree on Your comments, but have chosen to describe the Method and will present the results seperately in additional work. Thank you for improving the paper and adding valuable References. Specific feedback on comment are listed below. Comments and Suggestions for Authors General comments: In a context where sustainable urban drainage system are more and more implemented, to filtrate and infiltrate storm water, there is a need to control the charge of contaminants in such system in order to make sure that they do not allow contaminants to impact groundwater. In this frame, the paper proposes a methodology to map heavy metals in sustainable urban drainage system (SuDS), based on in situ measurement on topsoils with a portable XRF tool. The methodological approach is rather well described, but it should be reorganized to allow a better understanding. The results are rather disappointing as no maps are presented. Answer: We have chosen to only address the method in this paper. The results will be published in separate papers. This is to not make the papers too lengthy. I agree on the suggestion showing results in maps. However, this will be done in a follow-up paper. The section is sometimes difficult to follow: different pXRF tool used (difference), comparability with lab analysis not clear, applicability. It should be reorganized and completed. The beginning of conclusion looks like an abstract. – conclusion is altered after suggestion. The authors should focus on more general conclusions: interest of proposed methodology, recommendations. In general, the authors refers very much to “this” study”, suggesting a report extract. Answer: Point taken. Changed accordingly. Detailed comments: • Title: The title sounds in accordance with the objectives of the paper. Should nevertheless the abbreviation SuDS be used? Suggestion: “Quick in situ trace elements mapping for urban drainage system management: a methodological approach” Answer: since this method can be used for several SuDS we would like to keep the title Instead of heavy metal, I prefer the term trace elements (more general). Answer: changed to Potential Toxic Elements (PTE) • Introduction: It should be revised to end with the objectives of the paper. 1st and 2nd paragraphs writing ok. The interest of checking the trace elements content of topsoils in SuDS for management purposes should be explained before giving the objectives of the paper (I would suggest putting it in between 1st and 2nd paragraphs). The figure doesn’t appear necessary in the introduction part. It could be placed in the Material and method section to present the functioning of the swales and explain the sampling strategy. Avoid repetition (eg pXRF quick and cost efficient). Answer: Changes as suggested The authors could add more references on pXRF use, like: • Lemière B. (2018), Bruno Lemiere. A Review of pXRF (Field Portable X-ray Fluorescence) Applications for Applied Geochemistry. Journal of Geochemical Exploration, Elsevier, DOI: 10.1016/j.gexplo.2018.02.006 • RAMSEY M.H., & BOON K.A., 2012 - Can in situ geochemical measurements be more fit-for-purpose than those made ex situ? Applied Geochemistry 27, p. 969–976. • JEAN-SORO L., C. LE GUERN, B. BECHET, T. LEBEAU, M.F. RINGEARD, 2014 - Origin of trace elements in an urban garden in Nantes, France, Journal of Soils and Sediments, DOI 10.1007/s11368-014-0952-y. Answer: we would like to thank the reviewer for these very relevant references, they are added to the text. • Materials and methods The 2nd paragraph of the introduction part of the section should be moved to the general introduction (as context), or to the discussion, or to the conclusion. Thank you for this suggestion. This work focus on the method, which is established within some disciplines but new in water management and SuDS. Therefore, the amount of results presented here is limited. After review of several reviewers, we choose to keep the present structure of the paper, but will keep your suggestion in mind in further work with the results. The section should be reorganized. A suggestion of organization is given here after 1. Principle of sampling strategy 1. Functioning of swales 2. General sampling strategy (2.3.) 2. Quantification of trace elements 1. In situ measurements 2. Principle of pXRF (agragetion of 2.1. and beginning of 2.2) 3. Tools used Explain the difference between the tools • Analysed samples Preparation, With/without vegetation 1. Lab analysis 1. Preparation, ICP MS… 2. Study sites 1. Description 2. Sampling/measurements Precise here period of measurements => soils dry or wet, organic matter… 4. Interpretation method Comparison between in situ measurements and lab analysis Comparison to threshold values Mapping ? • Results It is very disappointing to see no map of trace elements for each site, whereas the title and objectives announce it. Answer: We have deliberately not included maps of results in this paper to focus on the method for mapping and not the results of the mapping. Even though pXRF is an established tool within geochemistry, it is a new tool within SuDS and water management for mapping of PTE. Professions for SuDS and water management is our aimed audience to improve their approach of maintenance. Although other studies have demonstrated the comparability of pXRF measurements and ICP/MS analyses, some matrix effect might disturb the correlation (influence of organic matter content, humidity of soil…, see eg. Lemiere et al, 2014). Answer: Yes, we agree and are aware of the matrix disturbance. However, for the purpose of the method the consideration are the national threshold values. These thresholds are far above disturbance limits. Will include in discussion to stronger point out the possible factors of disturbance of the results. The proposed figure do not make it easy to get convinced that the correlation is sufficient. How about a figure compiling the correlation between in situ measurements and lab analysis? Answer: Good point. This will be included in the results paper. For the data presented here, the number of data are not large enough to get a significant correlation. Previous reviewers have commented this as well. Lemiere, B., Laperche, V., Haouche, L. & Auger, P., 2014 - Portable XRF and wet materials: application to dredged contaminated sediments from waterways. Geochemistry: Exploration, Environment, Analysis 14, 257-264. Thank you for a good reference. It seems that the tests carried out in each site have given purposes=> reorganize the section according to these purposes 1. Comparability of pXRF and ICP/MS measurements – reliability of pXRF in situ measurements 2. Influence of pXRF tool and support of measurement (vegetation/topsoil) Test carried out on Heiloo site. Conclude on these aspects 3. Position of measurement profiles regarding inlet, outlet and other site specificities 4. Discussion 1. It should focus on the methodological aspects At the end of the discussion, the authors write: “The XRF quick scan analytical procedure could be used for water management and water quality control to comply with the Water Framework Directive [16] and the CIRIA‐The SuDS manual [2].” : one may understand that you can analyze water with pXRF. The sentence should be modified to avoid the possibility of such interpretation. Answer: Thank you for this feedback. Sentence is changed to clear misunderstanding. • Conclusion The beginning of conclusion looks like an abstract. The authors should focus on more general conclusions: interest of proposed methodology, recommendations for current sampling, as well as future sampling (survey). Answer: We have changes, according to suggestions. Adding remarks: How to ensure an efficient survey? Soil sampling conduct to soil removal. How to make sure that sampling nearby in the future (survey) will provide comparable results => need to verify short distance spatial variations of contents. Have you made such tests in situ? Answer: This will be included in follow up work that will focus on the results. • Figures and tables: Current figures difficult to read. The Figures should be revised to fit with the revised organization of results and discussion section. Fig 5: Size much too large (should fit in 1 p max). rem : Answer: Size of figure 5 is changed to fit one page. Size of fig 7 is convenient. Separate vegetation/topsoil results (support and in situ tool) from pXRF/ICP-MS results (measurement/analytical tool) to make results easier to understand. Dotted lines for vegetation will make reading easier too. Cu in profile 1, 2 and 3 are set to 0. This must be changed to detection limit. Answer: Cu below LOD commented in figure text Fig 6: too many lines, difficult to read. Same comments as for fig 5 (pXRF/ICP-MS comparison). Fig. too large too. Fig 7: what do the both profiles correspond to? Answer: The aim of the paper is to demonstrate the method for pXRF to map potential toxic elements in SuDS. The results are primarily shown to demonstrate the spatial distribution of the elements, and that the water flow controls the pattern of build-up. How about indicating threshold values in Fig 5 to 7? Answer: All results are well below intervention values, thereby no point in adding in the figures. Add at least maps comparing results to Dutch threshold values could be shown in addition (green dots below threshold value, yellow, red above intervention value. Comparison between pXRF and ICP/MS results should be shown. Answer: We have chosen to only address the method in this paper. The results will be published in separate papers. This is to not make the papers too lengthy. I agree on the suggestion showing results in maps. However, this will be done in a follow-up paper. • References: Well documented paper. Some suggestions nevertheless (see comments in the introduction and discussion sections) Answer to reviewer: Thank you for your feedback, that contributes to improve the paper. The title points out that As pointe

Reviewer 3 Report

The authors describe a quick scan mapping technology to detect and quantify the envromental pollutants i.e.  Potential Toxic Elements in three urban drainage sytems. Authors have employed a portabale X-ray fluorescence spectrometer. authors have also verified some of the results with ICP-MS which is a highly sensitive technique. The study is well designed, the experiments are well conducted and the manuscript is also well written. The mauscript can be accepted with minor formating and language correction as mentioned below

Page numbers start on page 2   (page 2 is marked as page 1)   

please replace "metre" with "m" in the text , for instance, in the section of materials  and methods. (1-2 m instead of 1-2 metres  and 5-7 m instead of 5-7 metres etc)

Page 1. Introduction. line 5. "ecosystem-based adaptation (EbA) and nature-based solutions (NBS) and more [3]) " need to be replaced by "ecosystem-based adaptation (EbA), nature-based solutions (NBS) and others [3]) a"

Page 1. Introduction. line 9. "can be incorporated management and maintenance" need to be replaced with "can be incorporated into management and maintenance"

Page 2. Introduction. line 4. "Studies by amongst others Jones " delete amongt others 

Page 2. Introduction. line 7-9. repharase the sentence "The high cost of soil analysis is the main reason the investigation of the environmental-technical functioning of rainwater facilities has not been systematically conducted on a large scale " 

Page 2. Introduction. line 11-12.  "Portable XRF measurements are an established " can be repalced with  "Portable XRF measurements provide an established "

Page 3, Materials and Methods.  "The proposed approach for mapping SuDS in used swales as pilot facilities " can be repalced with  "The proposed approach was exploited for mapping SuDS in used swales as pilot facilities"

Page 3, Materials and Methods.  "The in situ pXRF measurements are of topsoil (0-3 cm)" can be replaced with  "The in situ pXRF measurements of topsoil (0-3 cm) were performed"

Page 3, Materials and Methods.  "The three pilot locations presented her and...." can be replaced with "The three pilot locations presented here and...."

Page 4, section 2.2. "...produces a set of unique set characteristic ...." can be replaced with "...produces a set of unique characteristic ...."

Page 4, section 2.3. "Measurements were collected at a systematic interval; 1 metre intervals were used." can be replaced with "Measurements were performed at a systematic interval of 1 m."

Page 4, section 2.3. "Such measurements will provide the background value and weather any build-up contamination is present in the swale." can be replaced with "Such measurements will provide the background value and whether any build-up contamination is present in the swale."

Page 4, section 2.4. "The soil samples were collected of the topsoil" can be replaced with "The soil samples of the topsoil were collected"

 Page 4, section 2.4. "18 days at 26 Celsius (min26 ◦C–max 29 ◦C)," can be replaced with "18 days at 26 ◦C (min26 ◦C–max 29 ◦C),"

Page 5, section 2.4.  figure 2 "Two soil samples for ICP -MS analysis was collected" can be replaced with "Two soil samples for ICP -MS analysis were collected"

Page 5, section 2.4.  figure 3 "Two soil samples for ICP -MS analysis was collected" can be replaced with "Two soil samples for ICP -MS analysis were collected"

Page 7, section 2.5.1.   "Profile 1 (P1) is close to the south-eastern end of the swale, profile 2 (P2) is 10 metre from P1 and profile 3 (P3) is 40 metre from P1" can be replaced with "Profile 1 (P1) is located close to the south-eastern end of the swale.  Profile 2 (P2)   and  profile 3 (P3) are situated 10m and 40 m from P1"

Page 9, section 3.2, " Values are high close to the inlet and.." can be replaced with "Values are high near the inlet and.."

Page 10, Figure 6, " National intervention values for topsoil is respectively lead (Pb) 530 ppm, zinc (Zn) 720 ppm and copper (Cu) 190 ppm (Table 1)." can be replaced with "National intervention values for topsoil are  530 ppm,  720 ppm and 190 ppm for Pb, Zn and Cu, respectively (Table 1)."

Page 11. please revise the sentence  " The use of pXRF for in situ measuring in the field is increasing and as is evidence of the tool’s reliability, most notably Radu and Diamond [24], Bull, Brown and Turner [38], Turner and Solman [39], Rincheval, Cohen and Hemmings [28], Turner et al. [29,40] and Lemière et al. [41]."

Page 11. " This has been also pointed out by Kalnicky and Singhvi [11]." can be replaced with " This notion has also been pointed out by Kalnicky and Singhvi [11]."

Page 12. Conclsions, " IA methodology of in situ " can be replaced with " A methodology of in situ "

Author Response

Thank you for Your review of this article and for improving Our work. Thank you for your feedback on this manuscript. Your comments helped improve the paper. Here are the changes made in detail: Page numbers start on page 2 (page 2 is marked as page 1) please replace "metre" with "m" in the text , for instance, in the section of materials and methods. (1-2 m instead of 1-2 metres and 5-7 m instead of 5-7 metres etc) All “metre” are changed to m as suggested Page 1. Introduction. line 5. "ecosystem-based adaptation (EbA) and nature-based solutions (NBS) and more [3]) " need to be replaced by "ecosystem-based adaptation (EbA), nature-based solutions (NBS) and others [3]) a" Changed as suggested Page 1. Introduction. line 9. "can be incorporated management and maintenance" need to be replaced with "can be incorporated into management and maintenance" Changed as suggested Page 2. Introduction. line 4. "Studies by amongst others Jones " delete amongst others Changed as suggested Page 2. Introduction. line 7-9. repharase the sentence "The high cost of soil analysis is the main reason the investigation of the environmental-technical functioning of rainwater facilities has not been systematically conducted on a large scale " Changed to ‘Systematic, large scale investigation of environmental-technical rainwater facilities has not been conducted due to the high cost of soil analysis. ‘ Page 2. Introduction. line 11-12. "Portable XRF measurements are an established " can be replaced with "Portable XRF measurements provide an established " Changed as suggested Page 3, Materials and Methods. "The proposed approach for mapping SuDS in used swales as pilot facilities " can be replaced with "The proposed approach was exploited for mapping SuDS in used swales as pilot facilities" Changed as suggested Page 3, Materials and Methods. "The in situ pXRF measurements are of topsoil (0-3 cm)" can be replaced with "The in situ pXRF measurements of topsoil (0-3 cm) were performed" Changed as suggested Page 3, Materials and Methods. "The three pilot locations presented her and...." can be replaced with "The three pilot locations presented here and...." Changed as suggested Page 4, section 2.2. "...produces a set of unique set characteristic ...." can be replaced with "...produces a set of unique characteristic ...." Changed as suggested Page 4, section 2.3. "Measurements were collected at a systematic interval; 1 metre intervals were used." can be replaced with "Measurements were performed at a systematic interval of 1 m." Changed as suggested Page 4, section 2.3. "Such measurements will provide the background value and weather any build-up contamination is present in the swale." can be replaced with "Such measurements will provide the background value and whether any build-up contamination is present in the swale." Changed as suggested Page 4, section 2.4. "The soil samples were collected of the topsoil" can be replaced with "The soil samples of the topsoil were collected" Changed as suggested Page 4, section 2.4. "18 days at 26 Celsius (min26 ◦C–max 29 ◦C)," can be replaced with "18 days at 26 ◦C (min26 ◦C–max 29 ◦C)," Changed as suggested Page 5, section 2.4. figure 2 "Two soil samples for ICP -MS analysis was collected" can be replaced with "Two soil samples for ICP -MS analysis were collected" Changed as suggested Page 5, section 2.4. figure 3 "Two soil samples for ICP -MS analysis was collected" can be replaced with "Two soil samples for ICP -MS analysis were collected" Changed as suggested Page 7, section 2.5.1. "Profile 1 (P1) is close to the south-eastern end of the swale, profile 2 (P2) is 10 metre from P1 and profile 3 (P3) is 40 metre from P1" can be replaced with "Profile 1 (P1) is located close to the south-eastern end of the swale. Profile 2 (P2) and profile 3 (P3) are situated 10m and 40 m from P1" Changed as suggested Page 9, section 3.2, " Values are high close to the inlet and.." can be replaced with "Values are high near the inlet and.." Changed as suggested Page 10, Figure 6, " National intervention values for topsoil is respectively lead (Pb) 530 ppm, zinc (Zn) 720 ppm and copper (Cu) 190 ppm (Table 1)." can be replaced with "National intervention values for topsoil are 530 ppm, 720 ppm and 190 ppm for Pb, Zn and Cu, respectively (Table 1)." Changed to: Work by Radu and Diamond [24], Bull, Brown and Turner [38], Turner and Solman [39], Rincheval, Cohen and Hemmings [28], Turner et al. [29,40] and Lemière et al. [41] have documented the pXRF tool’s reliability. Page 11. please revise the sentence "The use of pXRF for in situ measuring in the field is increasing and as is evidence of the tool’s reliability, most notably Radu and Diamond [24], Bull, Brown and Turner [38], Turner and Solman [39], Rincheval, Cohen and Hemmings [28], Turner et al. [29,40] and Lemière et al. [41]." Sentence revised Page 11. " This has been also pointed out by Kalnicky and Singhvi [11]." can be replaced with " This notion has also been pointed out by Kalnicky and Singhvi [11]." Changed as suggested Page 12. Conclusions, " IA methodology of in situ " can be replaced with " A methodology of in situ " Changed as suggested

Round 2

Reviewer 1 Report

Major revisions asked, minor revisions made. Paper slightly improved nevertheless. Most of previous comments maintained (including méthods and results).

if this is a 2 parts paper, then clearly indicate it in the title, introduction and conclusion. 

Conclude clearly on methodological recommendations: not only pXRF use and what to do in case of PET contents above threshold values. The sampling strategy is also of importance.

Author Response

Major revisions asked, minor revisions made. Paper slightly improved nevertheless. Most of previous comments maintained (including méthods and results). if this is a 2 parts paper, then clearly indicate it in the title, introduction and conclusion. Conclude clearly on methodological recommendations: not only pXRF use and what to do in case of PET contents above threshold values. The sampling strategy is also of importance. Cover Letter – Sci – updated The article “Portable XRF Quick-Scan Mapping for Potential Toxic Elements Pollutants in Sustainable urban Drainage Systems: A Methodological Approach” is as the title points out, first and foremost, a methodological approach of mapping PET pollutants in SuDS such as swales. Water is a transporting media for pollutants, especially in urban areas. Within the field of sustainable urban drainage systems and water management, the build-up of pollutants is of both maintenance and financial interest. It cost time and money to do clean-up. This article addresses an easy, both time and cost-efficient method to do a quick mapping of potential toxic elements using the portable XRF instrument. As this is a methodology article we present results that demonstrate the suggested method. This work is part of a national study that is not finished and the results on a national level will be pub-lished on a later stage if budget and stakeholders allow this, but we cannot promise a timeframe for the publication. We are satisfied, as multiple reviewers that we present a paper which presents the new method as well as first results which the scientific community can benefit from. This manuscript ID (water-561244) has previously been submitted to Water with following major revisions after suggestions from three reviewers. The editor suggested that this manuscript better fit the scope of Sci than Water. Feedback and changes after review is available. In the review process of Sci, which agree that the topic of the article fits the scope of the journal, two reviewers have commented the article where one of the reviewers reported minor revisions. In this work we are in the junction between established disciplines. Methods well known in one discipline is used in another. To please all experts fully is not possible. Abstract: Sustainable urban drainage systems (SuDS) such as swales are designed to collect, store and infiltrate a large amount of surface runoff water during heavy rainfall. Stormwater is known to transport pollutants, such as particle-bound Potential Toxic Elements (PTE), which are known to often accumulate in the topsoil. A portable XRF instrument (pXRF) is used to provide in situ spatial characterization of soil pollutants, specifically lead (Pb), zink (Zn) and copper (Cu). The method uses pXRF measurements of PTE along profiles with set intervals (1 meter) to cover the swale with cross-sections, across the inlet, the deepest point and the outlet. Soil samples are collected, and the In-Situ measurements are verified by the results from laboratory analyses. Stormwater is here shown to be the transporting media for the pollutants, so it is of importance to investigate areas most prone to flooding and infiltration. This quick scan method is time and cost-efficient, easy to execute and the results are comparable to any known (inter)national threshold criteria for polluted soils. The results are of great importance for all stakeholders in cities that are involved in climate adaptation and implementing green infrastructure in urban areas. However, too little is still known about the long-term functioning of the soil-based SuDS facilities. Keywords: portable X-ray fluorescence spectrometer (pXRF); Potential Toxic Elements (PTE); lead (Pb), zinc (Zn); copper (Cu); topsoil; sustainable urban drainage systems; SuDS; LID; BMPs; WSUD; GI; SCMs

Reviewer 3 Report

This is a very good scientific piece of work which describes a quick scan mapping technology to detect and quantify the environmental pollutants i.e.  Potential Toxic Elements in three urban drainage systems. Authors have employed a portable X-ray fluorescence spectrometer. Authors have also verified some of the results with ICP-MS which is a highly sensitive technique. The study is well designed, the experiments are well conducted and the manuscript is also well written. I thank the authors for considering my comments and incorporating some of the changes. The manuscript can be accepted with minor formatting and language correction as mentioned below

I have found both ”meter” and ”metre” in the text. Please use only one spelling type

Page 1. Introduction. line 5. "ecosystem-based adaptation (EbA) and nature-based solutions (NBS) and more [3]) " need to be replaced by "ecosystem-based adaptation (EbA), nature-based solutions (NBS) and others [3]) a"

Page 1. Introduction. line 9. "can be incorporated management and maintenance" need to be replaced with "can be incorporated into management and maintenance"

Page 2. Introduction. line 4. Delete "amongst others" in sentence "Studies by amongst others Jones "

Page 2. Introduction. line 7-9.Please rephrase the sentence "The high cost of soil analysis is the main reason the investigation of the environmental-technical functioning of rainwater facilities has not been systematically conducted on a large scale " 

Page 2. Introduction. line 11-12.  "Portable XRF measurements are an established " can be replaced with  "Portable XRF measurements provide an established "

Page 3, Materials and Methods.  "The proposed approach for mapping SuDS in used swales as pilot facilities " can be replaced with  "The proposed approach was exploited for mapping SuDS in used swales as pilot facilities"

Page 3, Materials and Methods.  "The in situ pXRF measurements are of topsoil (0-3 cm)" can be replaced with  "The in situ pXRF measurements of topsoil (0-3 cm) were performed"

Page 3, Materials and Methods.  "The three pilot locations presented her and...." can be replaced with "The three pilot locations presented here and...."

Page 4, section 2.2. "...produces a set of unique set characteristic ...." can be replaced with "...produces a set of unique characteristic ...."

Page 4, section 2.3. "Such measurements will provide the background value and weather any build-up contamination is present in the swale." can be replaced with "Such measurements will provide the background value and whether any build-up contamination is present in the swale."

Page 4, section 2.4. "The soil samples were collected of the topsoil" can be replaced with "The soil samples of the topsoil were collected"

Page 5, section 2.4.  figure 2 and 3 "Two soil samples for ICP -MS analysis was collected" can be replaced with "Two soil samples for ICP -MS analysis were collected"

Page 10, Figure 6, " National intervention values for topsoil is respectively lead (Pb) 530 ppm, zinc (Zn) 720 ppm and copper (Cu) 190 ppm (Table 1)." can be replaced with "National intervention values for topsoil are  530 ppm,  720 ppm and 190 ppm for Pb, Zn and Cu, respectively (Table 1)."

Page 11. please revise the sentence  " The use of pXRF for in situ measuring in the field is increasing and as is evidence of the tool’s reliability, most notably Radu and Diamond [24], Bull, Brown and Turner [38], Turner and Solman [39], Rincheval, Cohen and Hemmings [28], Turner et al. [29,40] and Lemière et al. [41]."

Page 12. Conclusions, " IA methodology of in situ " can be replaced with " A methodology of in situ "

Author Response

Thank you for Your review of this paper. Sorry for all the stupid mistakes that were left in the manuscript. Changes are made according to Your suggestions. Page numbers start on page 2 (page 2 is marked as page 1) please replace "metre" with "m" in the text , for instance, in the section of materials and methods. (1-2 m instead of 1-2 metres and 5-7 m instead of 5-7 metres etc) All “metre” are changed to m as suggested Page 1. Introduction. line 5. "ecosystem-based adaptation (EbA) and nature-based solutions (NBS) and more [3]) " need to be replaced by "ecosystem-based adaptation (EbA), nature-based solutions (NBS) and others [3]) a" Changed as suggested Page 1. Introduction. line 9. "can be incorporated management and maintenance" need to be replaced with "can be incorporated into management and maintenance" Changed as suggested Page 2. Introduction. line 4. "Studies by amongst others Jones " delete amongst others Changed as suggested Page 2. Introduction. line 7-9. repharase the sentence "The high cost of soil analysis is the main reason the investigation of the environmental-technical functioning of rainwater facilities has not been systematically conducted on a large scale " Changed to ‘Systematic, large scale investigation of environmental-technical rainwater facilities has not been conducted due to the high cost of soil analysis. ‘ Page 2. Introduction. line 11-12. "Portable XRF measurements are an established " can be replaced with "Portable XRF measurements provide an established " Changed as suggested Page 3, Materials and Methods. "The proposed approach for mapping SuDS in used swales as pilot facilities " can be replaced with "The proposed approach was exploited for mapping SuDS in used swales as pilot facilities" Changed as suggested Page 3, Materials and Methods. "The in situ pXRF measurements are of topsoil (0-3 cm)" can be replaced with "The in situ pXRF measurements of topsoil (0-3 cm) were performed" Changed as suggested Page 3, Materials and Methods. "The three pilot locations presented her and...." can be replaced with "The three pilot locations presented here and...." Changed as suggested Page 4, section 2.2. "...produces a set of unique set characteristic ...." can be replaced with "...produces a set of unique characteristic ...." Changed as suggested Page 4, section 2.3. "Measurements were collected at a systematic interval; 1 metre intervals were used." can be replaced with "Measurements were performed at a systematic interval of 1 m." Changed as suggested Page 4, section 2.3. "Such measurements will provide the background value and weather any build-up contamination is present in the swale." can be replaced with "Such measurements will provide the background value and whether any build-up contamination is present in the swale." Changed as suggested Page 4, section 2.4. "The soil samples were collected of the topsoil" can be replaced with "The soil samples of the topsoil were collected" Changed as suggested Page 4, section 2.4. "18 days at 26 Celsius (min26 ◦C–max 29 ◦C)," can be replaced with "18 days at 26 ◦C (min26 ◦C–max 29 ◦C)," Changed as suggested Page 5, section 2.4. figure 2 "Two soil samples for ICP -MS analysis was collected" can be replaced with "Two soil samples for ICP -MS analysis were collected" Changed as suggested Page 5, section 2.4. figure 3 "Two soil samples for ICP -MS analysis was collected" can be replaced with "Two soil samples for ICP -MS analysis were collected" Changed as suggested Page 7, section 2.5.1. "Profile 1 (P1) is close to the south-eastern end of the swale, profile 2 (P2) is 10 metre from P1 and profile 3 (P3) is 40 metre from P1" can be replaced with "Profile 1 (P1) is located close to the south-eastern end of the swale. Profile 2 (P2) and profile 3 (P3) are situated 10m and 40 m from P1" Changed as suggested Page 9, section 3.2, " Values are high close to the inlet and.." can be replaced with "Values are high near the inlet and.." Changed as suggested Page 10, Figure 6, " National intervention values for topsoil is respectively lead (Pb) 530 ppm, zinc (Zn) 720 ppm and copper (Cu) 190 ppm (Table 1)." can be replaced with "National intervention values for topsoil are 530 ppm, 720 ppm and 190 ppm for Pb, Zn and Cu, respectively (Table 1)." Changed to: Work by Radu and Diamond [24], Bull, Brown and Turner [38], Turner and Solman [39], Rincheval, Cohen and Hemmings [28], Turner et al. [29,40] and Lemière et al. [41] have documented the pXRF tool’s reliability. Page 11. please revise the sentence "The use of pXRF for in situ measuring in the field is increasing and as is evidence of the tool’s reliability, most notably Radu and Diamond [24], Bull, Brown and Turner [38], Turner and Solman [39], Rincheval, Cohen and Hemmings [28], Turner et al. [29,40] and Lemière et al. [41]." Sentence revised Page 11. " This has been also pointed out by Kalnicky and Singhvi [11]." can be replaced with " This notion has also been pointed out by Kalnicky and Singhvi [11]." Changed as suggested Page 12. Conclusions, " IA methodology of in situ " can be replaced with " A methodology of in situ " Changed as suggested

Round 3

Reviewer 1 Report

I thank the authors for additional precisions. 

The transfer of methodology presented in this paper for the long-term management of SUDs is of great importance due to the recent large implementation of such water management systems and the operational applicability of the described method.

The authors propose a sampling strategy with pXRF, to allow a quick scan of PTE accumulation in topsoils of SUDs. They illustrate their proposal with the example of 3 swales. As they focus on methodological aspects (mainly sampling strategy with pXRF as a quick in situ/or lab analytical tool), few results are presented. The sampling strategy applied to SUDs can be seen as the innovative aspect of the paper. 

Although I believe some improvements could still be made in this interesting manuscript, it seems that the paper is of sufficient quality though to be published. I propose therefore only some minor suggestions. 

I am looking forward to discovering the forthcoming papers presenting more results.

Detailed comments: (yellow underlining in joined file)

  • Introduction

"...in cases where low detection levels are not required, time-consuming and costly laboratory analyses could be viewed as superfluous "=> "as almost superfluous" (need to carry out some lab analyses for calibration, ICP/MS generally carried out, especially if content close to threshold values).

  • Materials and methods2nd paragraph would rather fit in the discussion or conclusion sectionTable 2 : 2 columns titles appear similar
  • Results and discussionDiscussion should be placed in a sub-section (3.4) (2nd paragraph p 12)
  • Conclusion
  • p13"The instrument’s detection limits are well below national threshold values which makes this method suitable for the purpose of evaluating the toxicity of soil in SuDS" => I am not sure there is a direct link between detection limits of pXRF and evaluating soil toxicity => I suggest to replace "evaluation the toxicity of soil' by 'measuring PTE content in soils for management purposes' p13 "With this quick-scan method, traditional soil sampling and
    laboratory analyses become superfluous" => same remark as before (introduction).

Author Response

Thank you for improving this paper with useful comments and Insight. Introduction "...in cases where low detection levels are not required, time-consuming and costly laboratory analyses could be viewed as superfluous "=> "as almost superfluous" (need to carry out some lab analyses for calibration, ICP/MS generally carried out, especially if content close to threshold values). Changed to “…in cases where low detection levels are not required, time-consuming and costly laboratory analyses could be minimized to control samples.” Materials and methods2nd paragraph would rather fit in the discussion or conclusion Changed as suggested, moved to conclusion sectionTable 2 : 2 columns titles appear similar Yes we agree, but that is how the reference defines the metal limits. NMHSPE 2000 Circular on Target Values and Intervention Values for Soil Remediation. The Netherlands Ministry of Housing, Spatial Planning and the Environment, Amsterdam. 2000. Available online: https://www.esdat.net/Environmental%20Standards/Dutch/annexS_I2000Dutch%20Environmental%20Standards.pdf Results and discussion - Discussion should be placed in a sub-section (3.4) (2nd paragraph p 12) Changed as suggested Conclusion p13"The instrument’s detection limits are well below national threshold values which makes this method suitable for the purpose of evaluating the toxicity of soil in SuDS" => I am not sure there is a direct link between detection limits of pXRF and evaluating soil toxicity => I suggest to replace "evaluation the toxicity of soil' by 'measuring PTE content in soils for management purposes Changed as suggested ' p13 "With this quick-scan method, traditional soil sampling and laboratory analyses become superfluous" => same remark as before (introduction). Changed to: “With this quick-scan method, traditional soil sampling and laboratory analyses can be minimized to control samples”.

Reviewer 3 Report

This is a very interesting manuscript which could be published  

Round 4

Reviewer 1 Report

Dear authors, 

Thanks for sending the revised version with evidence of the changes made taking into account the remarks. Everything is fine for me now.

I wish to underline that some misunderstanding happened with the 1st revised version of the paper. The version I could download from the website corresponded actually to the initial version submitted, not to the modified one that I discovered after the 3rd review process. I do not know for which reason. Due to this, I had the feeling that no modification was made, explaining my answer. This conducted to the end some delay in the review process. Everything is OK now. 

I am now looking forward to the next paper presenting the results.

Best regards